# Searching for the heritage of the Second Sino-Japanese War: A study on the site selection strategy of the defence industrial buildings of the National Resources Commission (1937–1945)

**Yangjie Wu** [ORCID] *

School of Architecture and Art Design, Hunan University of Science and Technology, Xiangtan, Hunan, China

* 717117646@qq.com

**Citation:** Wu Y (2024) Searching for the heritage of the Second Sino-Japanese War: A study on the site selection strategy of the defence industrial buildings of the National Resources Commission (1937–1945). PLoS ONE 19(11): e0311436. https://doi.org/10.1371/journal.pone.0311436

**Data Availability Statement:** The data are available from Figshare. (https://doi.org/10.6084/m9.figshare.26798704.v1), and all relevant data are

## Abstract

China's heritage protection programme has gradually included the defence industry, and many industrial buildings related to defence have been included in the national list of cultural relics for protection. The relevant research mainly concerns the value of these heritage sites, lacking of research on the site selection and construction of buildings. This field also ignores the significant historical background of the Second Sino-Japanese War. This paper researches the National Resources Commission (NRC) during the Second Sino-Japanese War. It studies its site selection strategy for the 28 factories built by the organization in Hunan and Chongqing. First, the NRC-related historical data were sorted, including the list of factories built before and during the war, and their own historical construction database was established. Then, "Historical Position Finding - Site Feature Construction - Site Selection Factor Evaluation" is used to analyse each site. Finally, the site selection strategy of the NRC before and after the war was analysed. The innovation of this research is to put forward a research method combining field investigation and historical archives based on the perspective of institutional history. The study determined the spatial location of each plant and provided a direct basis for subsequent protection and utilization.

## 1. Introduction

The protection and utilization of defence industrial buildings is a crucial link to promote the inheritance of a history of military excellence. Since October 2019, when defence industrial buildings were selected for the first time in the "Eighth Batch of National Key Cultural Relic Protection Units" as a group of ten sites, relevant studies have been carried out. The Second Sino-Japanese War is an unavoidable topic in the history of the Chinese war, but the distribution and location of key defensive sites are not clear, and there is an urgent need to carry out basic research. In the study of modern Chinese architecture, the eight years between 1937 and 1945, when the war took place, have been regarded as the low point of architectural activity in

within the manuscript and its Supporting Information files.

**Funding:** This research was funded by the National Natural Science Foundation of China, a grant. Number 5197847. The funders had no role in study design, data collection and analysisdecision to publish, or preparation of the manuscript.

**Competing interests:** The authors have declared that no competing interests exist.

China [1, 2]. As a direct result of the war, urban construction was interrupted in the coastal region, which is where China's modernization began earlier and is also more urbanized. While the war destroyed infrastructure in some cities, it stimulated construction in other regions [3]. Due to the instinctive reaction to defend the country, the focus of national construction quickly shifted to the defence industry, stimulating engineering construction. With the gradual deepening of research, the NRC was included in research. Its wartime defence industrial system played a vital role in the war.

The NRC was the core of the national government's involvement in essential heavy industry and strategic national defence after the September 18th Incident. Industries under the agency's jurisdiction included electric power production, petroleum exploration, machinery and equipment manufacturing, chemical metallurgy, and the power alcohol industry. The widespread belief at the top of the National Government that Japan would soon launch a war of aggression against China was a key impetus for the rapid formation of the agency. In February 1932, the NRC's predecessor, the National Defence Planning Commission (NDPC), was secretly established. The NDPC's initial plan was to survey the nation's mineral, transportation, and human resources and use them to develop a preparedness plan. The NDPC name was considered too ostentatious by senior government officials, who specifically replaced the word "defence design" with "resources" in the name when the agency reorganized its internal operations [4]. After two internal reorganizations in 1935 and 1938, the NRC absorbed technical experts and investment projects related to industrial construction from other government departments and began to push forward with the construction of state-run heavy industries at full speed. From 1936–1938, the NRC built its defence industry primarily within Hunan. From 1938 to 1945, due to the war, the NRC quickly moved its ambitions to the five southwestern provinces of China, with the most projects located in Chongqing. By the end of the war, the NRC grew from a team of 31 technical experts to a ministerial-level government agency containing more than 200,000 technical personnel, operating in almost every industrial sector in the country [5].

This raises several research questions. Are there any NRC-related buildings left in Hunan and Chongqing? Site selection is not only an essential part of the architectural design process but also a critical step that affects subsequent operating costs and defence strategies. Against this particular historical background, how was the site selection considered? What were the specific plans? More importantly, are there any differences between prewar and wartime site selection strategies?

The NRC occupies a vital position in China's economic history, industrial development, and the history of state-owned enterprises. For this study, the NRC is used as the starting point mainly for the following advantages. First, the NRC had the most extensive management and construction scale, with 128 industrial and mining enterprises under its jurisdiction by December 1945. Second, the NRC had the broadest geographical scope, involving Hunan, Chongqing, Sichuan, Guizhou, Yunnan, Guangxi, and other southwest areas of China. Finally, the NRC covered the largest number of technical experts and operated in the most comprehensive industrial categories. Therefore, the in-depth analysis of the construction process of the NRC can not only effectively fill the lack of research on national defence industrial architecture but also expand the content of the chapter on wartime architecture in modern architectural history. In the context of the current research on urban renewal in China, research on industrial heritage has become increasingly fruitful. A series of research results have been formed around critical historical events such as the "156 Projects" [6, 7], the"Third-Front Construction" [8, 9], and Saving the "Three Materials" [10, 11]. Current industrial heritage research scholars focus less on wartime industrial areas in 1949 and before. It should not be ignored that this part of industrial development is scattered in the deep mountains across southwest China, and the situation for preservation is not optimistic. Investigation and research are urgently needed.

This paper researches the factories operated by the NRC in Hunan and Chongqing, adopting a combination of qualitative and quantitative methods to locate, identify, and record the industrial building remains. The site selection strategy for the factories is analysed, providing a direct basis for subsequent protection and reuse. The research of this paper always emphasizes the mutual accord among field investigation, historical maps, engineering archives, and historical documents, which have important theoretical and practical significance for the determination and protection of architectural remains in the country.

## 2. Literature review

### 2.1. NRC research trends and archives resources

It was not until the 1980s that NRC-related research became a hot spot. Before that, NRC history was a compilation of oral and written accounts of senior NRC staff [12]. Zheng, who used to work in the agency, conducted a detailed investigation of the establishment and development of the NRC [13]. Xue comprehensively reviewed the activities of the NRC in various historical periods [14]. Both works are based on detailed observations of documents in original archives and constitute significant overarching work in the field. Other studies focus on specific works of the NRC, including industrial activities [15, 16], factory and mine construction [17–19], industrial relocation [20, 21], state-owned enterprise systems [22–24], international cooperation on significant projects [25, 26], talent training [27, 28] and other aspects, presenting a complete process of institutional development. In the study of internationalization, William C. Kirby made full use of overseas archives to discuss the cooperation between the Chinese National government and Germany in industrial strategy, regime building, and social modernization, evaluated the industrial plan of the NRC, and argued for the need for continuity in the historical perspective of NRC construction [29]. Bian claims that the wartime state responded to the overall crisis, which led to the formation of the "Danwei" model of Chinese state-owned business, using the NRC factory as a representative case study [30]. Cheng analysed the origin, process, and results of the three industrial construction plans of the NRC and concluded that Chinese industrial plans after 1949 were directly affected by the NRC [31].

For the NRC's archival resources, the core materials are distributed in the Second Historical Archives of China in Nanjing and Academia Historica in Taipei, with 26,297 volumes and 27,985 volumes, respectively. Early Chinese historians used the Nanjing archives as their primary materials and the compilations of archival records published the Academia Historica as their auxiliary materials [32]. Academia Historica released its NRC-related holdings for the first time in 2019, which can be viewed online by logging in to the system. In addition to the relevant written materials, the public works archives, containing drawings and images, are the core historical materials of this study, including plant site plans, land acquisition drawings, survey reports, and general layouts. Other local archival information and books are also an essential reference for the research [33–35].

### 2.2. Industrial buildings during the Second Sino-Japanese War

The study of wartime industrial architecture lies at the intersection of industrial heritage and modern architecture. The research on China's industrial heritage began in 2006 and then was followed by a wide range of survey activities in major cities [36–38], gradually covering the industrial buildings of the war period. Most of the research on industrial heritage in Hunan is concentrated in Changsha, Zhuzhou, and Xiangtan. This approach not only ignores the NRC's industrial construction activities and architectural heritage in more rural areas of Hunan but also lacks a targeted investigation of the whole province. Comparatively speaking, Chongqing's rich historical archives and preserved factory ruins provide a basis for in-depth research, and

several findings have emerged [39–41]. In the study of modern architecture, the comprehensive results involve some wartime industrial areas in southwest China, but the content is more focused on civil architecture rather than industrial architecture [42, 43]. Although wartime industrial architecture has not yet formed an independent field, many scholars have been involved in researching architectural technology [44] and architect practices [45–47].

## 2.3. Research on site selection of modern industrial buildings in China

Location determination and site selection are essential links in industrial building construction. The research on industry against the background of the Second Sino-Japanese War mainly focused on the selection of locations for factories in coastal areas during the retreat. Modern historians have extensively collected primary data sources and investigated in detail the whole process of factory transfer to the southwest [48, 49]. In architecture, Xu Subin's research team has built a database of modern industrial buildings in China. They use GIS technology to elucidate the spatial evolution of modern industry [50], the spatial distribution of known industrial heritage sites [51], and the layout and site selection of factories in the early modern period [52], including wartime industrial buildings. In addition to considering the resources and traffic conditions, studies of wartime construction give substantial attention to the concealment and defensiveness of the building land [53]. Based on wartime urban defence, Xie conducted a historical study of Chongqing's air defence evacuation, industrial site selection, and urban management system [54].

The above studies have contributed to the research on wartime industrial buildings, but there are still some gaps. More specifically, the following questions remain:

Problem 1: The existing studies do not give enough attention to the industrial heritage of Hunan and Chongqing and ignore the use of critical historical materials from the NRC. If the engineering archives are fully used, these untapped resources can provide new ideas for researching industrial architecture in wartime.

Question 2: Although NRC research is beginning to extend to individual plants, it is still a historical overview in general. In particular, many studies lack field investigations and stay at the stage of historical data mining and document analysis. Therefore, fieldwork at heritage sites is urgently needed.

Question 3: The Second Sino-Japanese War is a complex and dynamic historical event that spans regions. The existing studies on wartime architecture mainly focus on static analysis, lacking a comparative study between prewar and wartime.

# 3. Study area and sample

## 3.1. Study area

Hunan and Chongqing are the main areas of this study and were the critical areas where the NRC made construction efforts before and during the war, respectively (Fig 1). Hunan is located in central China, with an area of 21.18 square kilometres, with outstanding advantages in railway and shipping access. It was the area of the main front of the confrontation between the military forces of China and Japan. Xuefeng Mountain bounds the landform of Hunan in the central part of the country, while the eastern part of Hunan is primarily flatlands, leaving the western region mountainous. Chongqing is located in southwest China, with an area of 8.24 square kilometres, located at the confluence of the Jialing River and the Yangtze River; it was the political and economic centre of the hinterland during the war. Chongqing is also

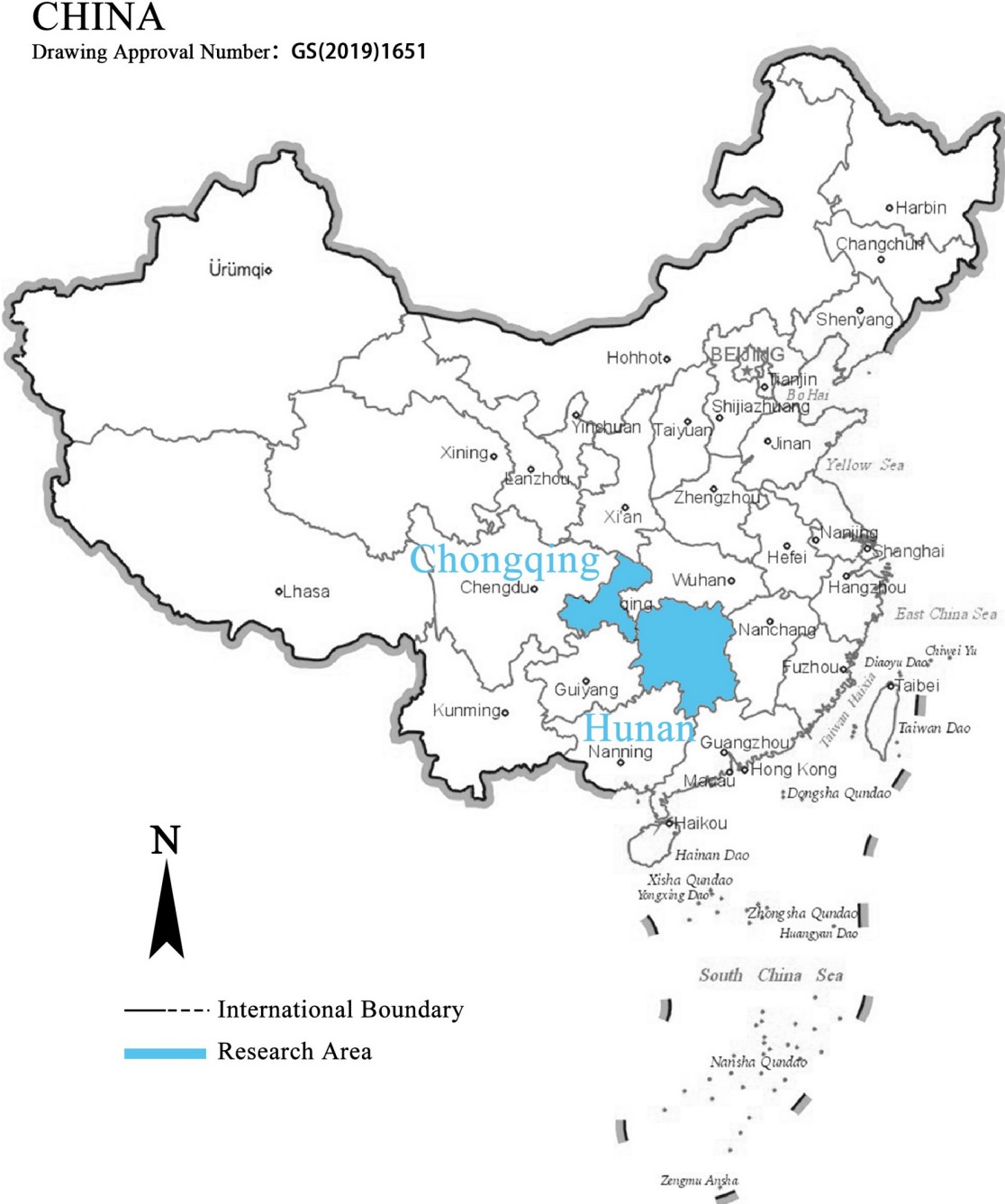

**Fig 1. Location of Hunan and Chongqing in China.** The base map images are from the China Standard Map Service System: http://bzdt.ch.mnr.gov.cn/.

known as a mountain city. The regional landform is mainly hilly, and the terrain is complex and changeable.

## 3.2. Sample organization

Before conducting fieldwork and data analysis, the observations for inclusion in the research sample must be determined. Sample selection is based on issues of the NRC bulletin. The

bulletin is issued monthly and details the operating conditions of the factories under the organization. After a detailed review, the Hunan and Chongqing regions contain 36 factories. The lack of historical information makes it impossible to locate these plants accurately, and plants that were obtained by NRC acquisition, rather than original construction, need to be eliminated. In the end, there are a total of 28 observations in the research sample. Detailed information is shown in the table, and the specific industries are divided into five categories: traditional industry (TI), smelting industry (SI), mining industry (MI), electrical industry (EI), and monopoly agency (MA). The pre-war and wartime separation points were set in August 1938 because, until then, Central China had been considered a safe zone (Table 1). After determining the research sample, the engineering archives of specific industrial plants and mines were analysed to assist with fieldwork and determine spatial locations (Fig 2).

## 4. Research processes

The whole research process is divided into four stages (Fig 3). The first stage is the data collection stage. For the sorted research samples, the historical and current data are sorted out,

**Table 1. Detailed information about the samples.**

| NO. | Name of Factory | Site | Time | Type |
|---|---|---|---|---|
| **Before the Second Sino-Japanese War (before August 1938)** | | | | |
| 1 | Central Electric Porcelain Works | Huangtuling, Changsha, Hunan | 1936 | TI |
| 2 | Central Radio Manufacturing Works | | 1936 | TI |
| 3 | Central Electrical Manufacturing Works | Xiashesi, Xiangtan County, Hunan | 1936 | TI |
| 4 | Central Iron and Steel Plant | | 1937 | SI |
| 5 | Central Machine Works | | 1936 | TI |
| 6 | Xiangtan Coal Mine Company | Xiangtan County, Hunan | 1937 | MI |
| 7 | Xiangjiang Electricity Works | Xiashesi, Xiangtan County, Hunan | 1937 | EI |
| 8 | Chaling Iron Mine Exploration Team | Chaling County, Hunan | 1936 | MI |
| 9 | Copper Refinery | Changsha, Hunan | 1936 | SI |
| 10 | Szechuan Petroleum Prospecting Corporation | Ba County, Chongqing | 1936 | MI |
| 11 | Antimony Administration | Lingling County, Hunan | 1936.1 | MA |
| **During the Second Sino-Japanese War (August 1938 - August 1945)** | | | | |
| 12 | Chenxi Coal Mine Company | Chenxi County, Hunan | 1938.8 | MI |
| 13 | Qiling Coal Mine Company | Qiling County, Hunan | 1940.4 | MI |
| 14 | Hunan Electric Company Changsha Branch | Changsha, Hunan | 1941.7 | EI |
| 15 | Hunan Electric Company Hengyang Branch | Hengyang, Hunan | 1942.9 | EI |
| 16 | Central Electric Porcelain Factory Hengyang Branch | | 1943.8 | TI |
| 17 | Xiangxi Electricity Works Yuanling Branch | Yuanling County, Hunan | 1939.1 | EI |
| 18 | Xiangxi Electricity Works Chenxi Branch | Chenxi County, Hunan | 1939.1 | EI |
| 19 | Tzeyu Steel Works | Ba County, Chongqing | 1944.4 | SI |
| 20 | Steel Plant Relocation Commission | | 1938.3 | SI |
| 21 | Electrochemical Refinery | Qijiang County, Chongqing | 1941.7 | SI |
| 22 | North Spring Alcohol Plant | Beibei County, Chongqing | 1941.5 | TI |
| 23 | Lung Chi Ho Waterpower Project Tao Hua Xi Branch | Changshou, Chongqing | 1937.7 | EI |
| 24 | Lung Chi Ho Waterpower Project Lower Tsing Yuan Tung Branch | | 1937.7 | EI |
| 25 | Wanxian Electricity Works | Wan County, Chongqing | 1938.8 | EI |
| 26 | Power Oil Plant | Tuwan, Chongqing | 1939.8 | TI |
| 27 | Refractory Materials Factory | Heishizi, Chongqing | 1941.10 | TI |
| 28 | Central Electrical Manufacturing Works Chongqing No. 4 Factory | Hualong Bridge, Chongqing | —— | TI |

Traditional Industry (TI), Smelting Industry (SI), Mining Industry (MI), Electric Industry (EI), Monopoly Agency (MA).

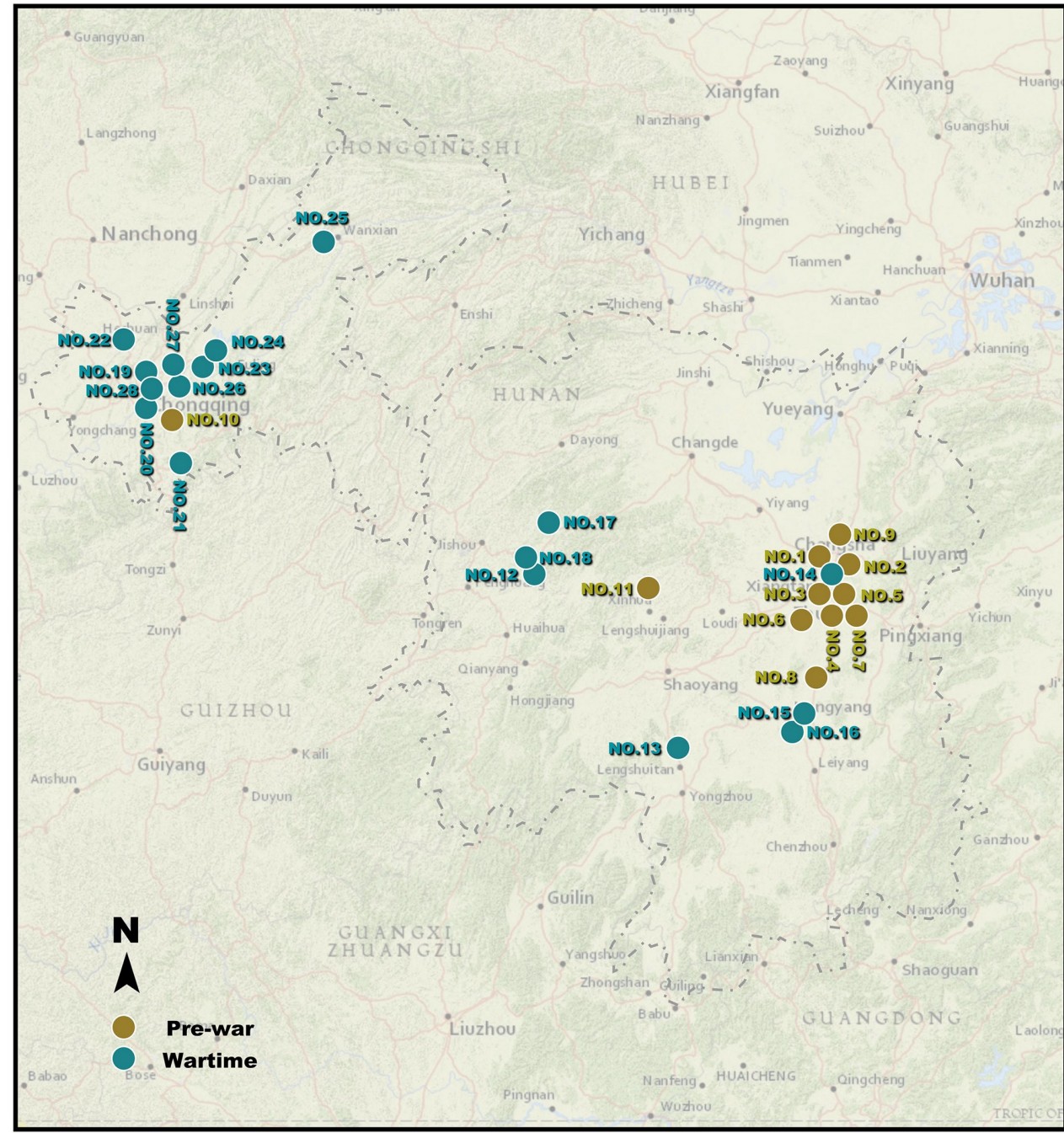

**Fig 2. Map of the NRC's 28 factories in Hunan and Chongqing.** The base data was from USGS EROS: https://www.usgs.gov/centers/eros. The yellow circle represents samples that before the war. The green circle represents samples that during the war.

mainly including historical documents, land expropriation plans, plant layout plans, field images and other materials. The second stage is the sample analysis stage, which mainly includes three aspects: historical location judgment, site feature construction and site selection element evaluation. The third stage is to study the results, focusing on the selection of single factor and the same industry for comparison and analysis. The final conclusion responds to the research questions and explores the NRC's pre-war and wartime site selection strategies.

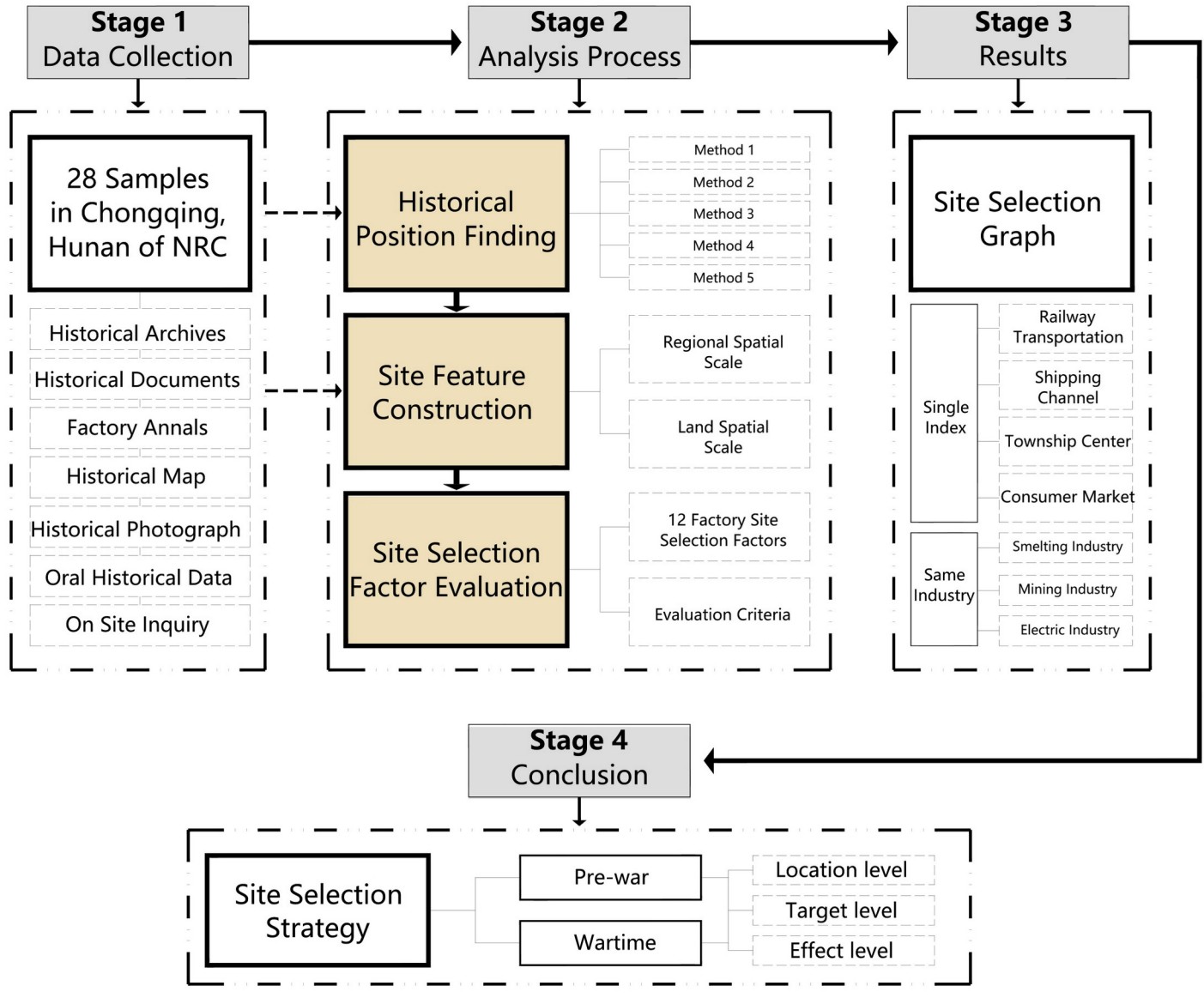

**Fig 3. Diagram of research method based on the three main research objectives.**

## 4.1. Analytical methods

The research process is shown as a flowchart. The research on site selection strategy uses a technical route that proceeds along the line "Historical Position Finding - Site Feature Construction - Site Selection Factor Evaluation," which includes three levels.

"Historical Spatial Positioning": Based on complete data, industrial remains can be accurately located on a microscale. In the course of fieldwork, it is not necessary to obtain additional work permits from relevant agencies. This is mainly due to the following reasons: After a large number of old monomer buildings were demolished, the land was transformed into urban construction land, so that each case was located in an open space; Some of the buildings are located outside the core site and do not require a survey permit.

"Site Feature Construction": the site features of modern urban maps that affect site selection are fully explored and restored. It includes the relationship between the factory location and

urban and rural areas, road and highway traffic routes, railway transit routes, shipping waterways, docks, waterway intersections, mountains, adjacent coal and iron energy distribution infrastructure, etc.

"Site Selection Factor Evaluation": the information on critical factors that affect the site selection of factories is explored and refined and an evaluation model is built with 12 factors.

In conclusion, the comparative analysis of the 28 cases summarizes the site selection strategy for defence industrial buildings. Most of the research observations are located in remote towns in southwest China, so it is necessary to collect historical maps and historical project archives. The historical maps mainly come from the long-term efforts of the research group in targeted collection and sorting, and the engineering archives mainly come from Academia Historica and Chongqing City Archives.

## 4.2. Historical position finding

Positioning is the initial step of site selection research and can narrow the scope of the building survey. To fully identify the historical space, based on the archival data and images, five methods of locating are developed. Different methods can be used simultaneously to increase the accuracy of spatial positioning (Fig 4).

Method 1: According to the NRC geological survey archives, text information about the village name, mining area name, and street name is extracted and directly matched to the site environment.

Method 2: The NRC's land allocation map, plant construction topographic map, and building configuration map are compared with the existing terrain to determine the construction site.

Method 3: In the face of incomplete construction files and the inability to obtain construction information, factory history, factory records, old photos, and other materials are analysed to infer the location of factory construction.

Method 4: In the field survey, the oral interview investigation is used to obtain circumstantial evidence to confirm and supplement the collected historical information.

Method 5: If there are still old buildings in the field research area, the factory's location can be directly confirmed.

## 4.3. Site feature construction

After determining the location of the factory, all kinds of spatial element information were sorted out, and the basic data of the sample were supplemented and verified by combining the results of on-site investigation. On the one hand, scaling down means the change from the macroscopic map perspective to the microscopic site perspective, and it is also the change from the two-dimensional plane to the three-dimensional picture. On the other hand, the combination of detailed environmental elements can enrich the cognitive picture of the site.

Based on the site conditions of each sample, the base map of the known historical space is drawn. The construction of site features is carried out at a regional scale and with land formations in scale (Fig 5). The regional base map is derived from the GIS platform, which can directly reflect the topography and river course with little change. Other historical environmental elements, such as city limits, rail lines, road routes, and power supply lines, must also

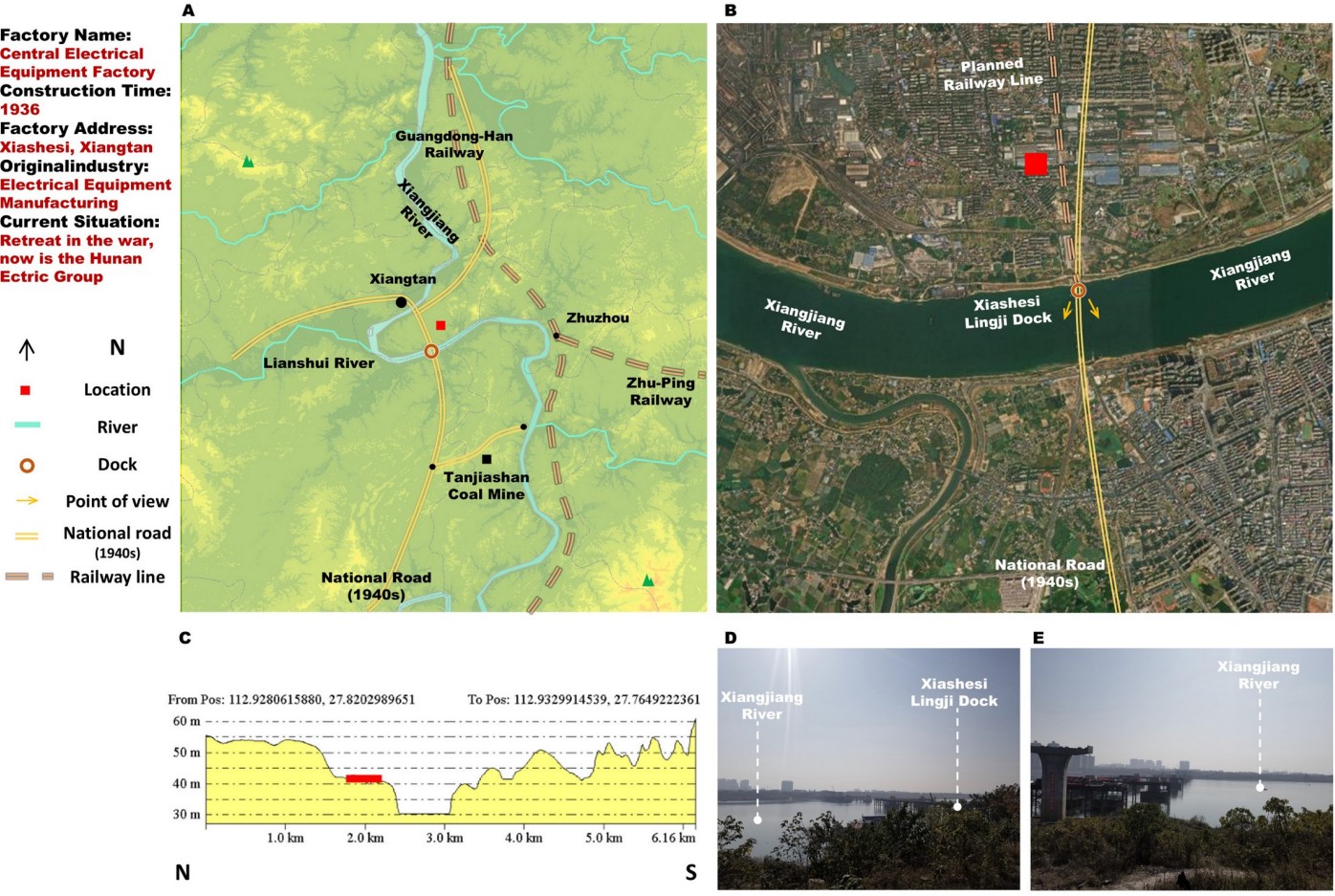

**Fig 4. Site features description of Central Electrical Manufacturing Works.** (A) Regional scale analysis image. The base maps were generated by ArcGIS 10.6 and were for illustrative purposes only. (B) Land satellite map. The base data was from USGS EROS: https://www.usgs.gov/centers/eros. (C) Site profile diagram. The base map were generated by 91 Weitu Assistant Software and were for illustrative purposes only. (D) and (E) Photo of the Xiashesi Dock. The base images were photographed by author.

be added to the map. Land formation analysis at scale is carried out with three sources: land satellite map, field image analysis, and site profile diagram. According to the requirements, the feature maps of 28 sample sites were drawn successively to form the drawing database.

## 4.4. Site Selection Factor Evaluation

"Site Selection Factor Evaluation" involves selecting influencing factors and constructing evaluation models. Taking 1937 as a temporal boundary, the site selection strategy was analysed for the periods before and during the war.

**4.4.1. Factory Site Selection Factors.** The following table lists several key factors that affect the location of a plant (Table 2). Identifying these factors mainly comes from the work reports and recollections of NRC construction staff [55]. The preliminary site selection report of each factory also mentioned some site selection factors, which were also included. These second-level factors include resources, traffic, production, and site considerations. The first three secondary indices can be obtained through the feature maps, and the analysis of the last site factor is mainly based on the project construction archives.

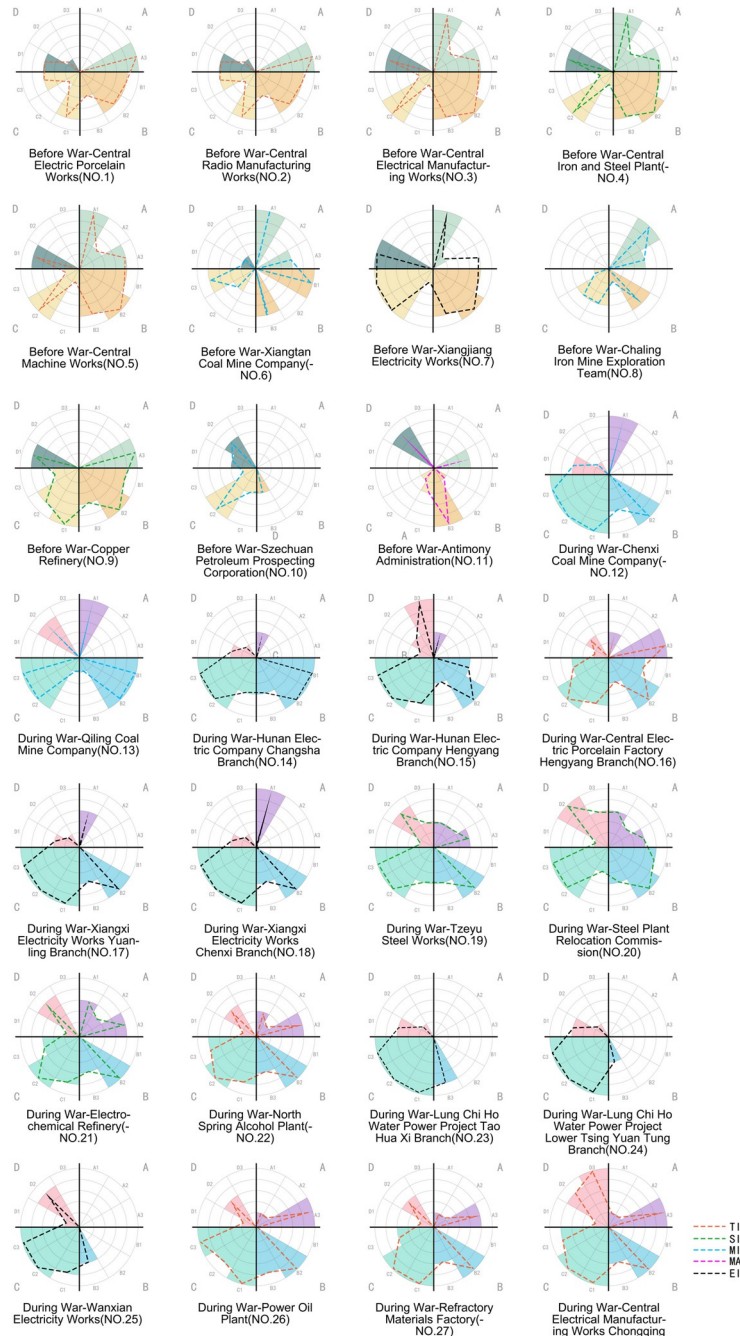

**Fig 5. Site selection evaluation radar map for 28 samples.**

**4.4.2. Site selection evaluation criteria.** There are 12 third-level factors, each with a score range of 0 to 5 points. The specific evaluation suggestions are shown in Table 3 (Table 3). Twenty engineers and researchers engaged in architectural design were invited to assign values to the diagrams of each site, calculate the average value of each data point, and draw radar maps according to the prewar and wartime conditions. Dotted lines of different colours distinguish factories in other industries.

**Table 2. Factory site selection factors.**

| Frist-level factor | Second-level factor | Third-level factor |
|---|---|---|
| Factory site selection of NRC | **A**<br>Resource factor | A1<br>Coal mine resources |
| | | A2<br>Iron ore resources |
| | | A3<br>Electric power resources |
| | **B**<br>Traffic factor | B1<br>Railway transportation |
| | | B2<br>Shipping channel |
| | | B3<br>Road transportation |
| | **C**<br>Production factor | C1<br>Township centre |
| | | C2<br>Water supply |
| | | C3<br>Consumer market |
| | **D**<br>Site factor | D1<br>Centralized layout |
| | | D2<br>Evacuation Arrangement |
| | | D3<br>Cave factory |

**Table 3. Evaluation criteria for factory site selection factors.**

| Third-level factor | Point scored suggestion |
|---|---|
| **A1** Coal mine resources | Distance to surrounding coal mines, iron mines, and power plants. Close proximity (5), relative proximity (4), average (3), relatively far (2), very far (1) and none (0). |
| **A2** Iron ore resources | |
| **A3** Electric power resources | |
| **B1** ailway transportation | Distance to the surrounding railway track. Railway direct (5), proximity (4), relative proximity (3), average (2 points), far away (1), none (0). |
| **B2** Shipping channel | Relation with shipping waterway: self-built harbour (5), Proximity (4), relative proximity (3), average (2), far away (1), none (0). |
| **B3** Road transportation | Highway facilities: highway through (5), Proximity (4), Relative proximity (3), Average (2), Far away (1), None (0). |
| **C1** Township centre | Distance to town centre, water source, and consumer market. Very close (5), relatively close (4), average (3), relatively far (2), very far (1) and none (0). |
| **C2** Water supply | |
| **C3** Consumer market | |
| **D1** Centralized layout | Concentration and compactness of surrounding factories. Very compact (5), compact (4), relatively compact (3), average (2), poor (1), none (0). |
| **D2** Evacuation Arrangement | Evacuation degree of factory buildings. Very evacuated (5), evacuation (4), comparative evacuation (3), average (2), poor (1), no or only single buildings (0). |
| **D3** Cave factory | The number of mountain cave factories in the factory. Very sufficient (5), sufficient (4), relatively sufficient (3), average (2), poor (1), none (0). |

## 5. Research on prewar and wartime site selection strategies for NRC

### 5.1. Overall evaluation

The overall result is shown in Fig 5. In general, the site selection of prewar factories is biased to favour access to resources and transportation. In contrast, the site selection of wartime factories is more concerned with transportation and production factors.

### 5.2. Single factor of site selection

Four factors are selected to observe the factory's location: railway transportation, shipping channels, township centre, and the consumer market, as shown in Fig 6.

**5.2.1. Railway transportation.** Analysing railway transportation factors shows that eight of the 11 prewar factories were along railway lines. Not connected with railway transportation are the Szechuan Petroleum Prospecting Corporation and Antimony Administration. They were typical feedstock-oriented factories. The former explored oil resources in the area of Chongqing Shiyougou village. At the same time, the latter was responsible for coordinating local antimony mining plants, which did not involve specific needs for production and transportation. During the war, only three of the 17 factories kept close contact with the railway: Qiling Coal Mine Company (Fig 7), Hunan Electric Company Changsha Branch, and the Steel Plant Relocation Commission. Among them, Qiling Coal Mine Company mainly provides power fuel for the Hunan-Guangxi Railway, and becomes an important refueling station in the rear. Railway transportation was a crucial concern in the prewar factory-siting process.

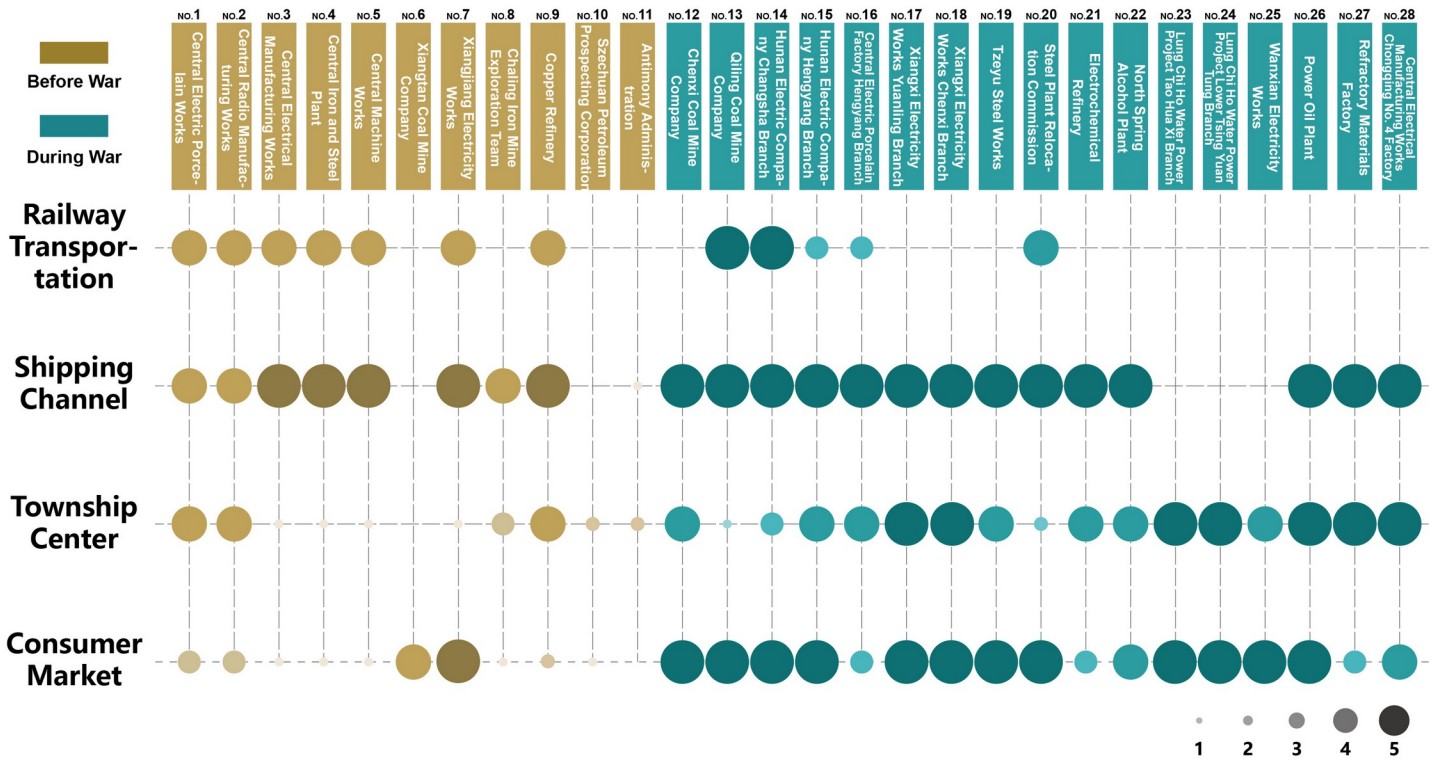

**Fig 6. Single factor score chart of site selection.**

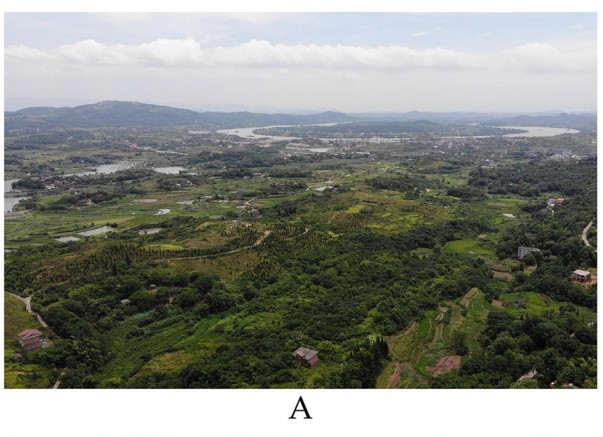

A

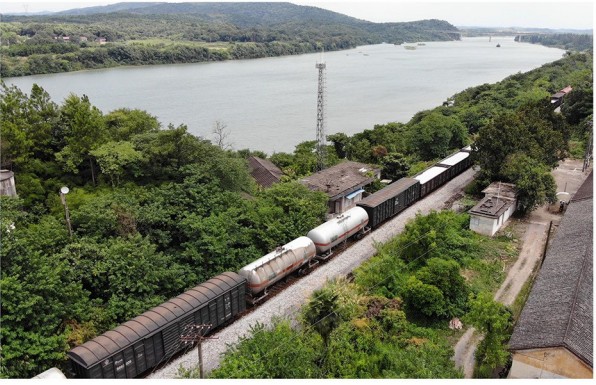

B

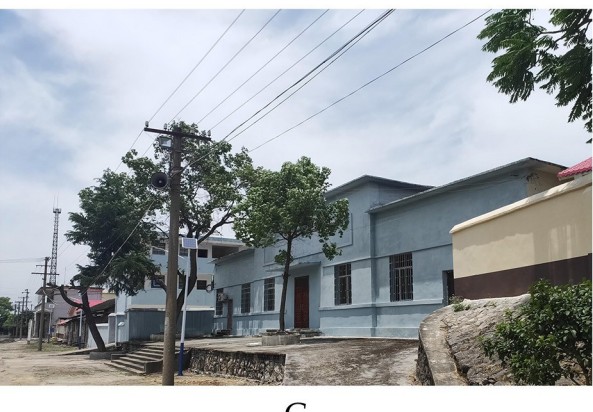

C

**Fig 7. On-site photos of Qiling Coal Mine Company.** The base images were photographed by author. (A) Aerial view of the coal mine site after natural greening; (B) Scene photo of Yijiaqiao station in Qiling Coal Mine Company; (C) Scene photo of Huangyangsi station in Qiling Coal Mine Company.

**5.2.2. Shipping channel.**   Excluding the three hydroelectric power stations, among the 25 factories, only the prewar Xiangtan Coal Mine Company, Szechuan Petroleum Prospecting Corporation (Fig 8), and Antimony Administration were not close to the water shore, because the above three samples are extremely dependent on the distribution of minerals. The factories built during the war were otherwise all on the waterway. Whether before or during the war, the shipping waterway had an essential influence on site selection. In addition, the proximity also makes it easy for the plant to obtain water for production and cooling.

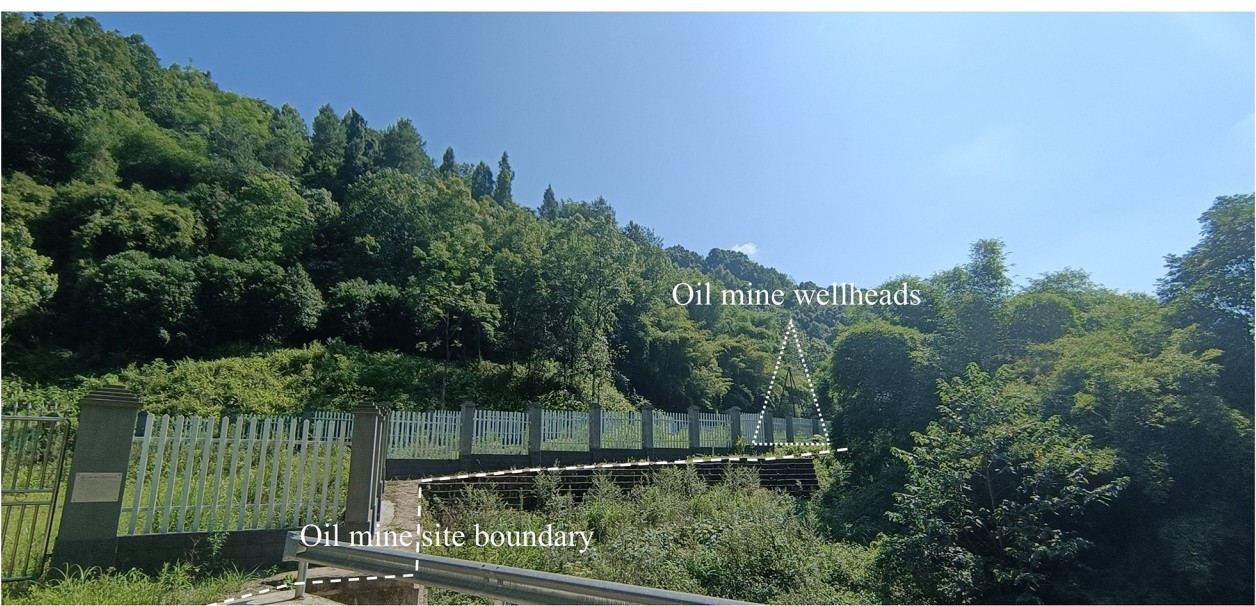

**Fig 8. Oil and gas well site of Szechuan Petroleum Prospecting Corporation.** The base image was photographed by author.

**5.2.3. Township centre.** Keeping a proper distance from urban areas and towns helps factories stay in touch with the outside world. Regarding the town centre, 11 prewar factories retained moderate contact with the city; in only 4 of 17 wartime factories was there distance from the city centre. Only the coal-mining companies and steel factories moved away from the city or town. For the township centre factor, the results show that only 4 of the 11 prewar factories maintained moderate contact with the town, and only the Qiling Coal Mine Company and Steel Plant Relocation Commission of the 17 wartime factories were far from town. There was a strong contrast between the prewar and the wartime decisions for siting NRC plants far away from existing towns. The prewar site selection paid more attention to the potential for future development and expansion of the land, and the wartime site selection strategy preferred the edge of town, which was put into production and used immediately.

**5.2.4. Consumer market.** In the relevant consumer market, there is a specific difference in the tendencies for prewar and wartime factory locations. Before the war, only Xiangjiang Electricity Works and Xiangtan Coal Mine Company were close to the consumption terminal. After the war, almost all factories were built adjacent to consumption areas. Especially for thermal power plants, power loss increases with distance. Therefore, the Yuanling branch plant and Chenxi Branch plant of the Xiangxi Electricity Works were located across the Yuanjiang River in the city (Fig 4).

## 5.3. Analysis of site selection in the same industry

In the TI category, four factories were opened before the war and five during the war. In terms of construction scale, the factory gradually changed from "large and specialized" prewar functions to "small and sophisticated" wartime functions. As reflected in the location, prewar factories sought high-quality land with rail, shipping, and road links, while wartime factories only needed land that could provide shipping traffic. Of course, wartime land is more focused on concealment and defence.

The war stimulated NRC construction. Before the war, there was only one power plant, but during the war, there were seven thermal and hydraulic power plants. Although the production cost of traditional thermal power plants is higher than that of hydropower plants, the terrain does not constrain the location, and the degree of freedom is relatively higher. Therefore, the site of thermal power plants in wartime is closer to the raw material supply and the power terminal, which minimizes the transportation costs and increases the power utilization efficiency. The Xiangxi Electricity Works Chenxi Branch, for example, is located in Tongwan Creek on the opposite bank of Chenxi County, at the intersection of the Yuanjiang River and Chenshui River. The primary purpose of selecting Tongwan Creek is to meet the electricity demand of the factories on the other side. Second, Chenxi County is rich in coal resources, which can meet the power plant's power generation needs in a short time. Therefore, the NRC chose the site to situate the plant near coal mines to facilitate fuel supply (Fig 9).

## 6. Discussion: Site selection strategy for the NRC

The first is to answer the question about the preservation of the remains of industrial buildings. Hunan and Chongqing, as the key areas of NRC during the Anti-Japanese War, are not very optimistic about the preservation of factory remains in these two areas. In the field

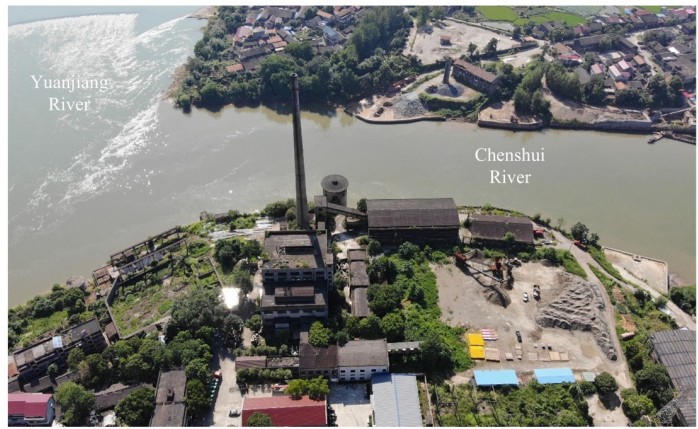

A

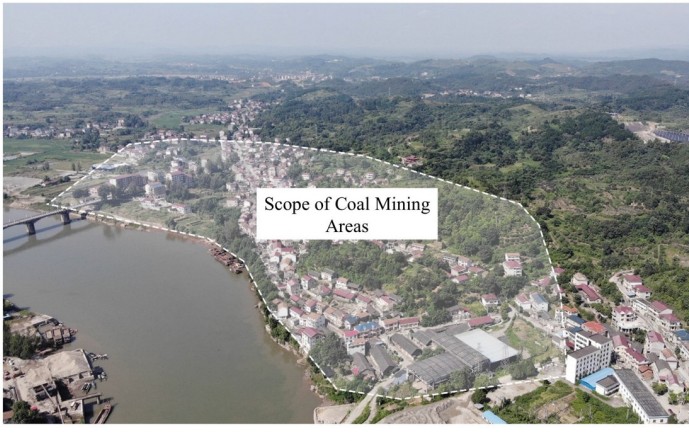

B

**Fig 9. Survey photos of Xiangxi Electricity Works Chenxi Branch.** The base images were photographed by author. (A) A power plant rebuilt on its original site;(B) Aerial view of the mining area.

investigation, it was found that few industrial remains were completely preserved, and architectural remains could be found in the samples of hydraulic power stations and later developed into large-scale industrial plants, such as Lung Chi Ho Waterpower Project and Central Electrical Manufacturing Works. A small number of examples, such as the Xiangjiang Electricity Works, continue the war-era industry, but the buildings have long since been demolished. A large number of samples located on the outskirts of the city no longer exist, and there are no corresponding markers in the site to awaken historical memories.

Secondly, based on 28 sample sites, combined with the engineering archives for each site, the principles of the NRC location strategy can be obtained. The site selection strategy is analysed from three strategic factors: location, goal, and effect. The location includes the macroscopic geographical location, the traffic situation on land, and the spatial relationship with mineral resources. The goal is mainly directed at factory construction and the fundamental problems to be solved. The effect is the expected long-term positive impact on the surrounding area after the completion of plant construction.

### 6.1. Site selection strategy before war

Keeping a distance from the coastline and having good traffic conditions are the basic requirements of NRC prewar site selection. The operations, including imports and exports of materials, are dominated by railway traffic and assisted by waterway shipping, and the two work together to complete transportation. The convenient Canton-Hankou railway and the Xiangjiang waterway meet the needs of NRC construction. Minerals are a resource guarantee ensuring continued factory production, and the coal and iron ore in Xiangtan met most industrial requirements.

For the prewar site targets, the main goal was to find new deposits of coal, iron ore, and nonferrous metal in a system that had independent development and operations, to facilitate an independent and autonomous capability to supply resources. In addition, due to German technical assistance, site selection also needed to be approved by foreign experts. In May 1936, the NRC invited German steel industry expert Durr to Hunan to determine the location of the central steel plant [35].

According to the strategic deployment, the three major factories were completed and put into operation in succession, effectively supporting the preparation of the coastal areas against Japanese invasion (Fig 10). The large-scale constructions of the NRC also attracted other government departments to locate key industries in Hunan. Neighbouring Zhuzhou began to develop the national defence and military sector in full, including the Artillery Technical Research Department organized by the Ministry of Military and Political Affairs, which began trial production of artillery shells. The Ministry of Railways also prepared to build a railway equipment machinery factory for the future construction of railways. According to the plan, when the establishment of national factories was completed, industries in the coastal areas would naturally be imported and transferred to the mainland interior. With a large amount of flat land, Xiashesi also met the needs of future industrial land expansion (Fig 11). The site selection strategy of the NRC before the war is shown in Fig 12.

### 6.2. Site selection strategy during the war

Safety in wartime factories was critical. The southwest area relied more on negative air defence during the war, so the excavation of shelters and cave factories gradually became increasingly common (Fig 13). Due to the lack of railway transportation in southwest China, factories could only choose to be close to the sea. At the same time, the construction area also needed to maintain a particular connection with sources of coal and iron production to facilitate the

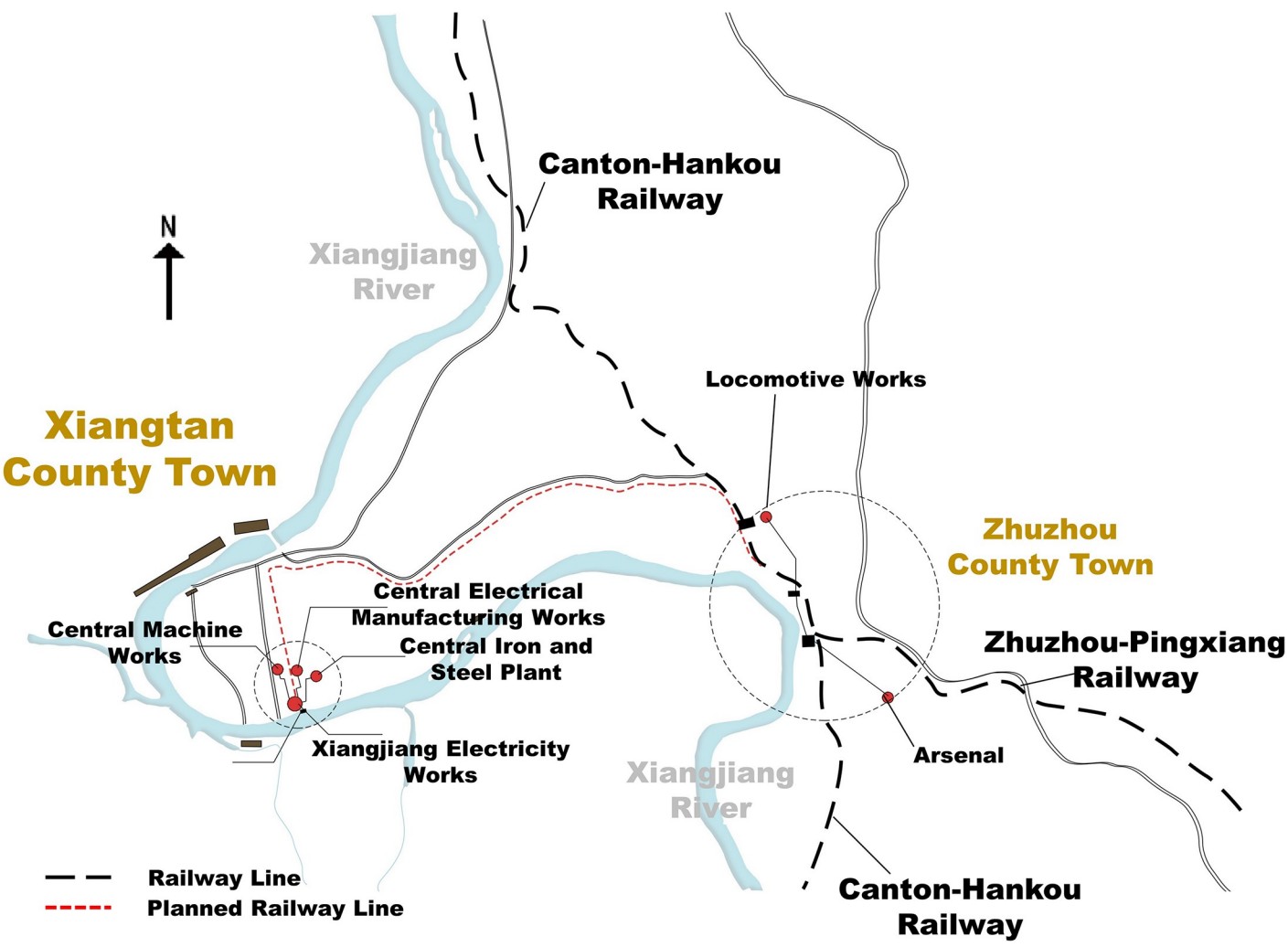

**Fig 10. Location map of three factories and power plants in Xiashesi, Xiangtan.**

timely supply of energy. It was equally important to maintain contact with the township, which provided daily operational support, and the consumer market, which processes the produced goods on time.

Auxiliary arsenal production is another objective of the site. In the process of the large-scale shift west of industrial production, the munitions factory prioritized deploying resources. It immediately returned to production, and the NRC provided continuous support in metal smelting and electricity. In the case of an external blockade, the NRC used state-owned capital to build factories that would connect upstream and downstream industries and relied on the wartime economy to accelerate the flow of materials, which can not only facilitate the production linkage between defence industry factories but also support private factories (Fig 14).

## 7. Conclusions

In the face of insufficient basic research, scarce remains of research objects, and incomplete historical data, this research focuses on the industrial relics of the Second Sino-Japanese War in China's inland urban and rural environments. It developed a research strategy that followed the technical route of "Historical Position Finding - Site Feature Construction - Site Selection

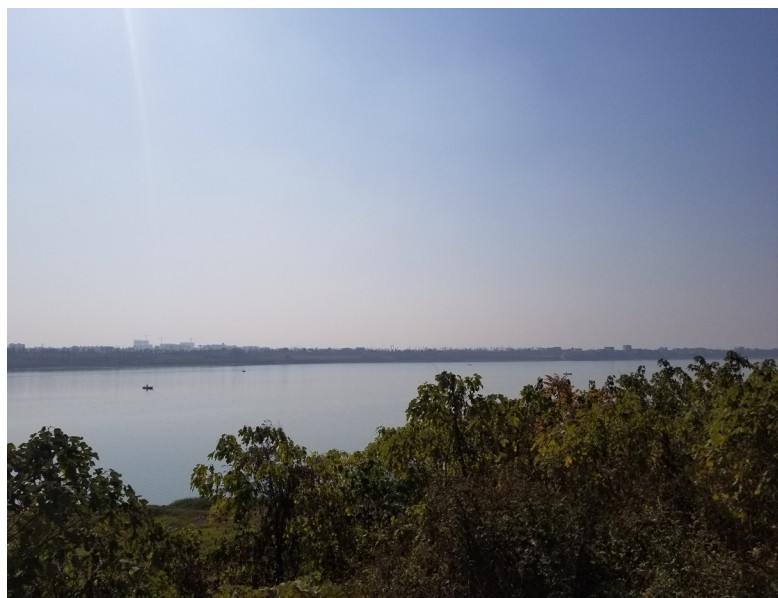
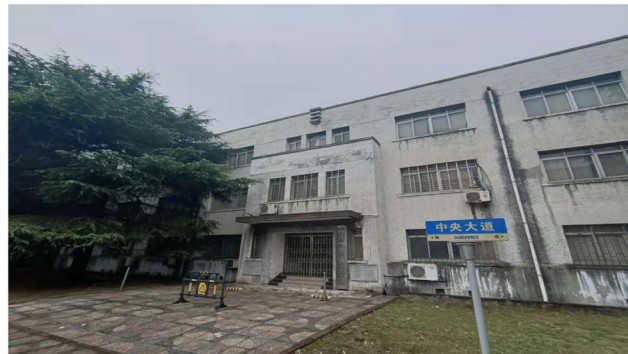
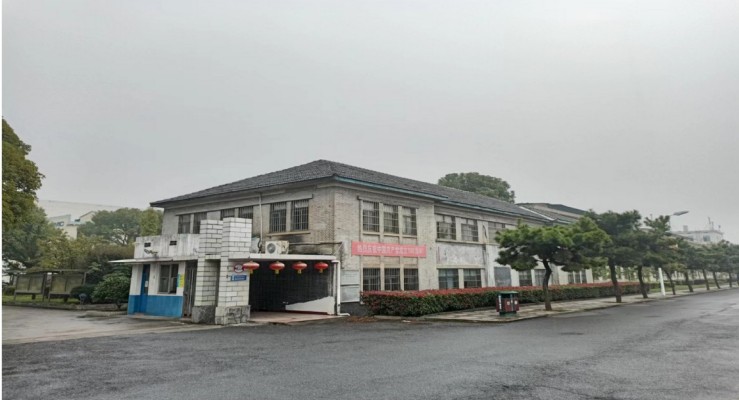
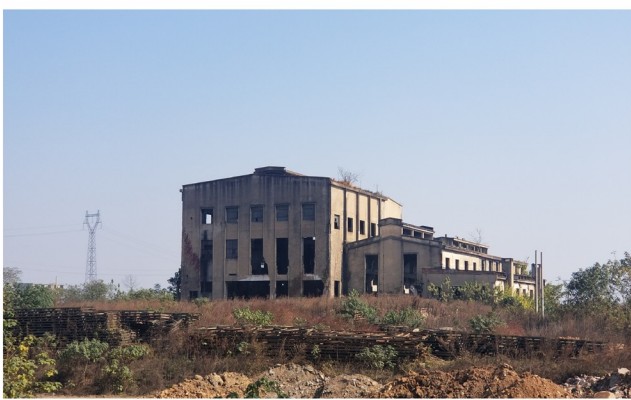

**Fig 11. Photo of existing historical buildings in Xiashesi, Xiangtan.** The base images were photographed by author. (A) Xiangjiang River Xiashesi wharf; (B) the old main office building of Central Electrical Manufacturing Works;(C) the old office building of Central Electrical Manufacturing Works; (D) the old site of Xiangjiang Electricity Works.

Factor Evaluation" to analyse site selection on an evaluative map of historical sites. Finally, the paper analysed the site selection strategy of defence industrial buildings through three objectives of "location, target, and effect".

Through the field investigation in Hunan and Chongqing, it is found that there are still some remnants of defence industrial buildings in urban areas and towns, and these buildings have not been fully excavated and protected. Field investigation found that although a small number of buildings have been preserved due to the continuation of the industry, and this part is also listed as a provincial cultural relic protection unit, a large number of architectural remains have died out with the expansion of the city, lacking the mark of historical memory. In the case that the material form of the city cannot be changed, the virtual reality scene can be considered to restore the original industrial production scene and fully activate the site.

Subsequent research shows that the site selection of NRC factories before the war was based on constructing a modern industrial base with high standards, three-dimensional transportation, and diversified cooperation. During the war, site selection strategies were based on

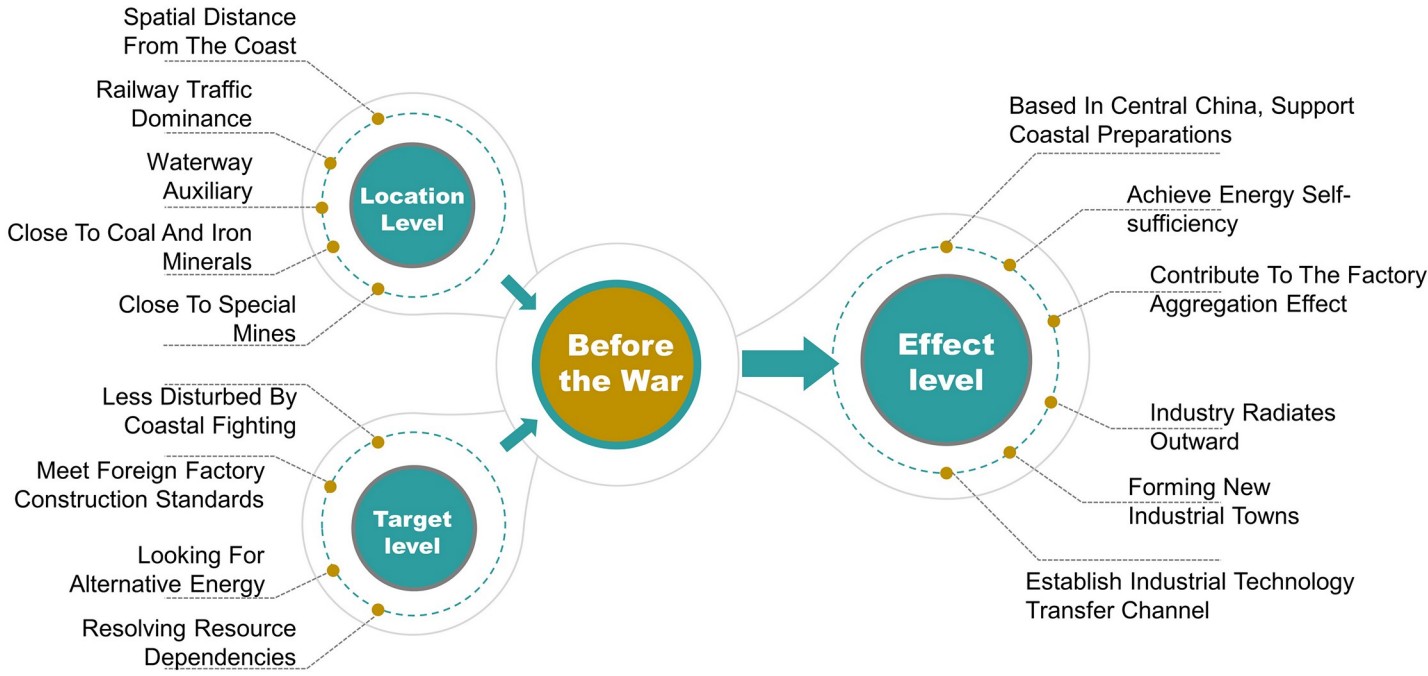

**Fig 12. Illustration of site selection strategy before the war.**

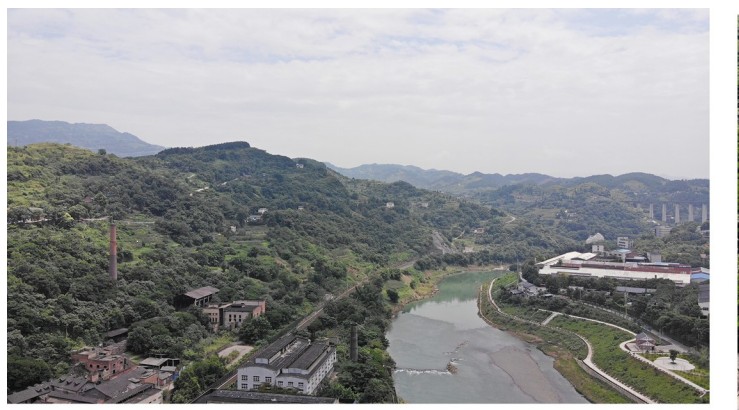

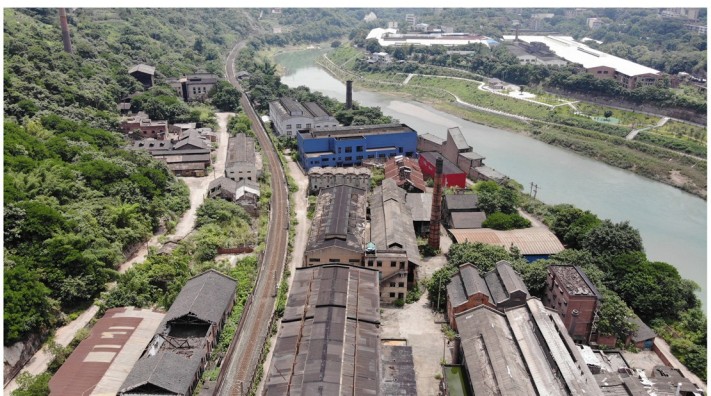

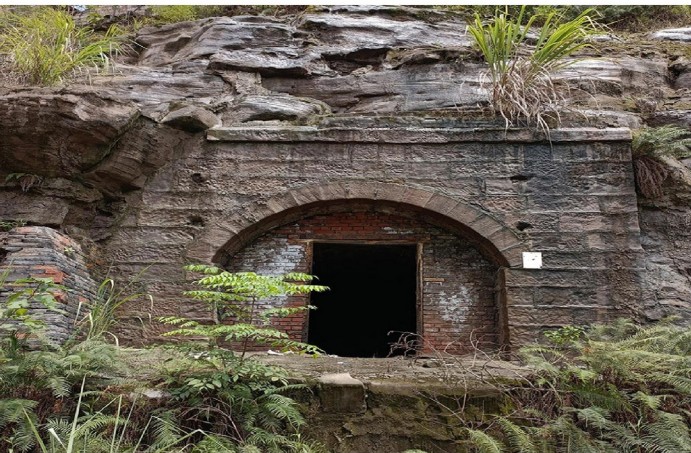

**Fig 13. Electrochemical Refinery site survey photos.** The base images were photographed by author. (A) Factories with their backs to the mountain facing the waterfront; (B) Abandoned factory buildings; (C) Air-raid shelter in Electrochemical Refinery. Source: photographed by author.

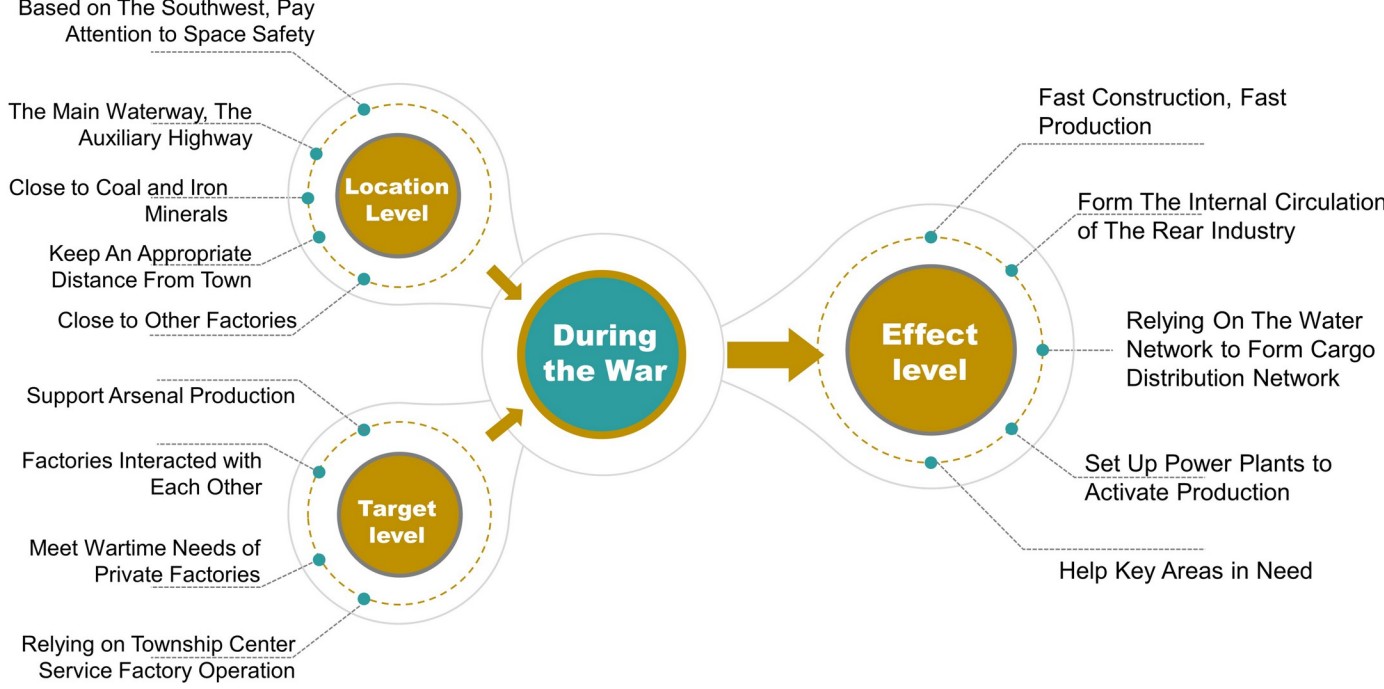

**Fig 14. Illustration of site selection strategy during the war.**

building a temporary production space with concealability, waterway traffic assistance, and industrial circulation.

This research guides the study and protection of the architectural heritage of the defence industry during wartime. At the level of the heritage unit, the study located, identified, and recorded architectural remains, which serve as the direct basis for heritage protection. In the built environment, the site selection strategy of the factory was studied and analysed, which enriched the understanding of the heritage to provide a reference for historical preservation. From the perspective of institutional history, this paper proposes a technical framework for the study of heritage and its built environment using field investigations, historical maps, engineering archives, and historical documents. Based on qualitative and quantitative research methods, it not only helps to reveal the historical and cultural value of heritage but also supplements research into modern architectural history and industrial heritage and provides a scientific basis for the protection of industrial buildings key to national defence.

## Acknowledgments

Thanks to Zhu Xiaoming for her help in this study, and to Zhang Zhenxin and Zhao Yanni for their help in the field investigation.

## Author Contributions

**Conceptualization:** Yangjie Wu.

**Data curation:** Yangjie Wu.

**Formal analysis:** Yangjie Wu.

**Funding acquisition:** Yangjie Wu.

**Investigation:** Yangjie Wu.

**Methodology:** Yangjie Wu.

**Project administration:** Yangjie Wu.

**Resources:** Yangjie Wu.

**Software:** Yangjie Wu.

**Supervision:** Yangjie Wu.

**Validation:** Yangjie Wu.

**Visualization:** Yangjie Wu.

**Writing – original draft:** Yangjie Wu.

**Writing – review & editing:** Yangjie Wu.

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
