## [Decision Letter · Decision Letter 0]

24 Jul 2024

PONE-D-24-15627Searching for the Heritage of the Second Sino-Japanese War: A Study on the Site Selection Strategy of the Defence Industrial Buildings of the National Resources Commission (1937-1945)PLOS ONE

Dear Dr. yangjie,

Thank you for submitting your manuscript to PLOS ONE. After careful consideration, we feel that it has merit but does not fully meet PLOS ONE’s publication criteria as it currently stands. Therefore, we invite you to submit a revised version of the manuscript that addresses the points raised during the review process.

We look forward to receiving your revised manuscript.

Kind regards,

Juha-Antti Lamberg, Ph.D.

Academic Editor

PLOS ONE

Journal Requirements:

4. We note that your Data Availability Statement is currently as follows: [If the data are all contained within the manuscript and/or Supporting Information files, enter the following: All relevant data are within the manuscript and its Supporting Information files.]

5. We note that [Figures 1,2,4,5,8,10,11 and 14] in your submission contain [map/satellite] images which may be copyrighted. All PLOS content is published under the Creative Commons Attribution License (CC BY 4.0), which means that the manuscript, images, and Supporting Information files will be freely available online, and any third party is permitted to access, download, copy, distribute, and use these materials in any way, even commercially, with proper attribution. For these reasons, we cannot publish previously copyrighted maps or satellite images created using proprietary data, such as Google software (Google Maps, Street View, and Earth). For more information, see our copyright guidelines: http://journals.plos.org/plosone/s/licenses-and-copyright.

a. You may seek permission from the original copyright holder of Figures 1,2,4,5,8,10,11 and 14 to publish the content specifically under the CC BY 4.0 license.  

Additional Editor Comments:

Thank you for the interesting piece of research focused on industrial archeology of Chinese military industry. The Reviewer has evaluated your research as appropriate and I agree: in your research field this is a solid piece of research and I support its publishing. Please follow the instructions of Reviewer 1 and re-submit following PLOS procedure.

Reviewers' comments:

Reviewer's Responses to Questions

**Comments to the Author**

1. Is the manuscript technically sound, and do the data support the conclusions?

Reviewer #1: Yes

2. Has the statistical analysis been performed appropriately and rigorously? 

Reviewer #1: N/A

3. Have the authors made all data underlying the findings in their manuscript fully available?

Reviewer #1: Yes

4. Is the manuscript presented in an intelligible fashion and written in standard English?

Reviewer #1: Yes

5. Review Comments to the Author

Reviewer #1: THANK YOU FOR AN INTERESTING PAPER. AS THE PAPER CONTAINS A LOT OF COMPLEX FIGURES, I WOULD PUT MORE EFFORT ON OPENING UP THESE FIGURES IN THE TEXT. THAT WOULD MAKE THE FIGURE MORE UNDERSTANDABLE TO THE READER. WHAT IS MORE, I WOULD FOCUS THE DISCUSSION SECTION OF THE PAPER MORE ON THE CONCRETE ANSWERS TO THE RESEARCH QUESTIONS THAT THE PAPER POSED.

6. PLOS authors have the option to publish the peer review history of their article (what does this mean?). If published, this will include your full peer review and any attached files.

Reviewer #1: No

---

## [Author Response · Author response to Decision Letter 0]

28 Aug 2024

Responses to the comments from editors and reviewers

On behalf of all the contributing authors, I would like to express our sincere appreciation of your letter and reviewers’ constructive comments concerning our article entitled “Searching for the Heritage of the Second Sino-Japanese War: A Study on the Site Selection Strategy of the Defence Industrial Buildings of the National Resources Commission (1937-1945)” (Manuscript PONE-D-24-1562). 

These comments are all valuable and helpful for improving our article. According to the reviewer's comments, we have made some modifications to our manuscript. We have studied them carefully and the revised contents are highlighted in ORANGE colour in the revised manuscript (Manuscript-R1). All reviewers’ comments are responded to below point by point. If there are any other modifications we could make, we would like very much to modify them and we really appreciate your help.

R1: Thank you for your reminder. We have reviewed the manuscript style again to ensure it conforms to PLOS ONE’s style requirements. We also refer to papers previously published in journals to ensure that the draft meets the requirements of the journal.

R2: Thank you for your reminder. In section 4.1, Analytical methods, the field research situation is explained, and the reasons for not requiring an institutional work permit are expounded. Detailed modifications are also provided in the revised manuscript as following:

“In the course of fieldwork, it is not necessary to obtain additional work permits from relevant agencies. This is mainly due to the following reasons: After a large number of old monomer buildings were demolished, the land was transformed into urban construction land, so that each case was located in an open space; Some of the buildings are located outside the core site and do not require a survey permit.”

R3: Many thanks to the editors for their comments on the funding information. I explain the funding situation in the "Funding Statement" section at the back of the manuscript as following:

“This research was funded by the National Natural Science Foundation of China, a grant. Number 5197847.”

4. Please confirm at this time whether or not your submission contains all raw data required to replicate the results of your study. Authors must share the “minimal data set” for their submission.

R4: Thanks to the editors for their comments on the research data. The minimum data of the study is the evaluation score of each sample. 

The authors provide the research data in the form of attachments. This attachment is a data sheet about the evaluation score of 28 samples. See the "Supporting Information" section of the manuscript. 

5. We note that [Figures 1,2,4,5,8,10,11 and 14] in your submission contain [map/satellite] images which may be copyrighted.

R5: Thank you for your reminder. In accordance with your suggestions, I have changed all the images of maps or satellite images based on USGS database and China Open Standard Map database. Some images with copyright disputes were deleted. It is worth reminding that the on-site photos in the manuscript are taken by myself, and there is no copyright problem.

6. Please review your reference list to ensure that it is complete and correct. If you have cited papers that have been retracted, please include the rationale for doing so in the manuscript text, or remove these references and replace them with relevant current references.

R6: Thank you for your reminder. The author reviewed all the references in the manuscript and found that there was no retraction of the cited papers.

7. reviewer #1: thank you for an interesting paper. As the paper contains a lot of complex figures, I would put more effort on opening up these figures in the text. That would make the figure more understandable to the reader.

R7: Thanks for the reviewer’s suggestion and we think this is an excellent suggestion. We have added a significantly more detailed description of the above questions to make the figure more understandable. Detailed modifications are also provided in the revised manuscript as following:

“The pre-war and wartime separation points were set in August 1938 because, until then, Central China had been considered a safe zone.” (Line 204 to 205 in Section 4)

“The whole research process is divided into four stages (Fig 3). The first stage is the data collection stage. For the sorted research samples, the historical and current data are sorted out, mainly including historical documents, land expropriation plans, plant layout plans, field images and other materials. The second stage is the sample analysis stage, which mainly includes three aspects: historical location judgment, site feature construction and site selection element evaluation. The third stage is to study the results, focusing on the selection of single factor and the same industry for comparison and analysis. The final conclusion responds to the research questions and explores the NRC's pre-war and wartime site selection strategies.” (Line 213 to 222 in Section 4)

“After determining the location of the factory, all kinds of spatial element information were sorted out, and the basic data of the sample were supplemented and verified by combining the results of on-site investigation. On the one hand, scaling down means the change from the macroscopic map perspective to the microscopic site perspective, and it is also the change from the two-dimensional plane to the three-dimensional picture. On the other hand, the combination of detailed environmental elements can enrich the cognitive picture of the site.” (Line 270 to 276 in Section 4.3.)

“Among them, Qiling Coal Mine Company mainly provides power fuel for the Hunan-Guangxi Railway, and becomes an important refueling station in the rear.” (Line 338 to 340 in Section 5.2.1.)

“Excluding the three hydroelectric power stations, among the 25 factories, only the prewar Xiangtan Coal Mine Company, Szechuan Petroleum Prospecting Corporation (Fig 8), and Antimony Administration were not close to the water shore, because the above three samples are extremely dependent on the distribution of minerals.” (Line 349 to 352 in Section 5.2.2.)

8. What is more, I would focus the discussion section of the paper more on the concrete answers to the research questions that the paper posed.

R8: Many thanks to the reviewers for the helpful comment. We have made changes to the manuscript and detailed modifications are also provided in the revised manuscript as following:

“The first is to answer the question about the preservation of the remains of industrial buildings. Hunan and Chongqing, as the key areas of NRC during the Anti-Japanese War, are not very optimistic about the preservation of factory remains in these two areas. In the field investigation, it was found that few industrial remains were completely preserved, and architectural remains could be found in the samples of hydraulic power stations and later developed into large-scale industrial plants, such as Lung Chi Ho Waterpower Project and Central Electrical Manufacturing Works. A small number of examples, such as the Xiangjiang Electricity Works, continue the war-era industry, but the buildings have long since been demolished. A large number of samples located on the outskirts of the city no longer exist, and there are no corresponding markers in the site to awaken historical memories.” (Line 407 to 417 in Section 6.)

“Through the field investigation in Hunan and Chongqing, it is found that there are still some remnants of defence industrial buildings in urban areas and towns, and these buildings have not been fully excavated and protected. Field investigation found that although a small number of buildings have been preserved due to the continuation of the industry, and this part is also listed as a provincial cultural relic protection unit, a large number of architectural remains have died out with the expansion of the city, lacking the mark of historical memory. In the case that the material form of the city cannot be changed, the virtual reality scene can be considered to restore the original industrial production scene and fully activate the site.” (Line 494 to 502 in Section 5.2.2.)

The author tried our best to improve the manuscript and made some changes. We appreciate for Editors/Reviewers’ warm work earnestly, and hope the correction will meet with approval. Once again, thank you very much for your comments and suggestions.

---

## [Editor Report · Decision Letter 1]

19 Sep 2024

Searching for the Heritage of the Second Sino-Japanese War: A Study on the Site Selection Strategy of the Defence Industrial Buildings of the National Resources Commission (1937-1945)

PONE-D-24-15627R1

Dear Dr. yangjie,

We’re pleased to inform you that your manuscript has been judged scientifically suitable for publication and will be formally accepted for publication once it meets all outstanding technical requirements.

Kind regards,

Juha-Antti Lamberg, Ph.D.

Academic Editor

PLOS ONE
---

## [Editor Report · Acceptance letter]

26 Sep 2024

PONE-D-24-15627R1 

PLOS ONE

Dear Dr. Wu, 

I'm pleased to inform you that your manuscript has been deemed suitable for publication in PLOS ONE. Congratulations! Your manuscript is now being handed over to our production team.

Kind regards, 

on behalf of

Dr. Juha-Antti Lamberg 

Academic Editor

PLOS ONE